# Spatiotemporal distribution characteristics of Nanjing place names—Based on data mining of Tang-Song poetry and online travelogues

**Weiya Zhang**[1], **Zhicheng Lai**[2], **Shu Tang**[1]*

**1** Faculty of Humanities and Social Sciences, Jinling Institute of Technology, Nanjing, China, **2** School of Geography, NanJing Normal University, Nanjing, China

* ts@jit.edu.cn

## Abstract

Tang-Song poetry, a distinguished element of China's traditional cultural heritage, is intricately linked with the historical and cultural development of Chinese cities. This paper uses Nanjing as a case study and applies digital humanities techniques to analyze and compare the spatiotemporal distribution of place names found in Tang-Song poetry with those in online travel narratives. The aim is to uncover key factors that have influenced the cultural continuity of these historical cities and their relevance today. Findings indicate that: (1) Locations mentioned in Tang and Song poetry show significant spatial differentiation, with urban areas displaying a clustered distribution and suburbs showing scattered hotspots. (2) The number of locations referenced in Song poetry increased significantly compared to Tang poetry, suggesting that Nanjing's economic growth heightened the city's appeal and inspired more literary output. (3) Song Dynasty poetry reflects a shift toward more neutral and negative emotions, with a marked decrease in positive expressions. This rise in negative sentiment can be traced to the decline in national strength from the Tang to the Song Dynasty, amplifying Nanjing's role as a place of reflection and mourning. (4) Nanjing's cultural hotspots, such as Xuanwu Lake and the Zhongshan Scenic Area, feature prominently in both Tang-Song poetry and modern travelogues. This study contributes to research in literary geography and literary tourism at the urban spatial level, offering fresh insights into the cultural legacy of historical cities.

## 1. Introduction

Guided by principles of new cultural geography, the geographic information and spatial significance embedded in classical Chinese literature have attracted considerable scholarly interest. Research shows that examining classical literature through a geographical lens not only enriches its interpretive depth but also enhances appreciation for its artistic conception [1]. Classical poetry, an essential component of Chinese literary heritage, reflects changes in the geographic environment and embodies local cultural imagery. Through vivid depictions of natural and cultural landscapes, as well as a genuine expression of thoughts and emotions, poets, embodying the dual identity of a litterateur and a traveler, have established a foundation for future generations in spacial landscape design and historical ambiance creation

**Data availability statement:** All relevant data are within the article and its Supporting information files.

**Funding:** This work was supported by the Ministry of Education Humanities and Social Sciences Foundation Project (Grant No. 24YJA790090).

**Competing interests:** The authors have declared that no competing interests exist.

during tourism development. Traditional humanities research on Chinese classical poetry has often prioritized temporal analysis, qualitative over quantitative methods, and individual case studies over collective examinations. In contrast, digital humanities technologies—characterized by a comprehensive set of data acquisition, analysis, and sharing techniques such as digital technologies (e.g., scanning, photography), digital management technologies (e.g., text encoding, semantic search), data analysis technologies (e.g., text analysis, geographic spatial analysis), data visualization technologies (e.g., knowledge mapping, scene simulation), and emerging tools like virtual reality (VR) and artificial intelligence (AI)—can explore the spatial distribution of classical poets' movements and emotions and the geographical distribution of poetic works across regions and cities from spatial, quantitative, and group perspectives [2]. This approach provides valuable insights for the construction of modern cultural landscapes and the strategic planning of urban tourism [3].

China's historical and cultural cities are vividly portrayed in classical poetry, making poetry a vital source for exploring the cultural imagery, historical contexts, and spatial representations of these cities. Classical poetry offers a lens through which we can understand the cultural boundaries and spatial imagery associated with these locations. However, existing research on the interplay between cities and classical poetry has largely focused on the temporal and spatial distribution of poets, often overlooking comprehensive studies across regions and periods that analyze poets' aesthetic preferences and perceptions of nature, history, and culture. Few studies employ text mining to extract historical spatial data over extensive regions and time scales to explore geographical distribution and spatial evolution. This gap limits analysis from both historical and spatial perspectives, impeding a deeper understanding of how cultural landscapes evolved [4]. Additionally, the limited use of digital humanities analytical methods has hindered the reconstruction of cultural landscape patterns and context image characteristics that reflect the residents' habitation, production, and lifestyle within the city. As a result, there is insufficient analysis of the formation of humanistic and natural landscapes in China's historical cities, the effective shaping and reuse of local literary landscapes, and their potential contributions to tourism planning through the development of culturally enriched landscapes.

Poetry often mirrors the author's perspectives, capturing their emotional responses to particular places. A poet's expression of emotions tied to a location thus becomes a vital indicator of the "people-place" relationship in geographical studies [5]. Emotion plays a central role in classical poetry, making an exploration of the spatial, temporal, and emotional shifts in classical poets' works a valuable approach for advancing knowledge in the humanities [6].

In summary, there is a scarcity of literature that employs digital humanities analytical methods and engages in comparative research between ancient and contemporary texts (classical poetry and modern travelogues) to uncover the historical reasons and emotional values behind the formation of cultural tourism attractions in historical and cultural cities. This paper uses Nanjing as a typical case study of a historical and cultural city, investigating how a city rich in literary resources can promote the development of heritage tourism by examining the evolving spatial patterns of tourism landscapes. Furthermore, this study offers valuable insights for literary tourism cities worldwide on how to explore the historical significance of literary landscapes and identify tourism development pathways from the perspectives of temporal evolution and comparisons between ancient and modern travelers.

## 2. Literature review

### 2.1 Research on literary geography

Literary geography is an interdisciplinary field exploring the relationship between literature and place, integrating both literary and geographical perspectives to examine relevant

phenomena and issues. The geographical approach emphasizes the influence of geographic elements on literary works, whereas the literary approach considers how these works represent and shape perceptions of place [7].

**2.1.1 Research from the literary perspective.** Scholars in literary studies primarily focus on three core areas. The first explores how natural geographical environments shape literary works and their authors, examining regional culture and literature through origins, cultural landscapes [8], and the movement patterns of literary figures. In terms of geographic distribution, researchers have observed that, influenced by political, economic, cultural, and geographic factors, the center of gravity (CoG) of literary figures in China has shifted across historical periods. During the Zhou, Qin, Western Han, Eastern Han, Three Kingdoms, Western Jin, Sui, and Tang dynasties, literary figures were concentrated in northern China (middle and lower reaches of the Yellow River). In contrast, during the Eastern Jin, Northern and Southern Dynasties, Song, Liao, Jin, Yuan, Ming, and Qing dynasties, literary figures were more prevalent in southern China (middle and lower reaches of the Yangtze River) [9]. Literary geography also involves the study of place names, drawing from literary sources and supported by geographical records from various dynasties [1]. As a key component of literary geography, the literary-geographic landscape exerts a distinct influence on literary creation, the thematic content of works, and their artistic style and character [10].

The second area of study explores the conceptualization and significance of literary works depicting natural landscapes. In literary geography, geographic space and landscape are central concepts. Literary works often feature both real and imagined geographic spaces, with many place names serving as dual symbols of geographic and imagery spaces, contributing to cultural representation and construction [11]. The geographic space in classical novels can be analyzed across three dimensions: inner versus outer space, narrative versus metaphorical space, and textual versus graphic space [12].

The third area of focus examines the relationship between literary genres and geography. This includes analyzing geographic culture and literature through the lenses of social class [13], the North-South divide [14], and groups of poets and genres [9]. The unique island environment of the UK and its maritime culture have significantly influenced the narratives, character traits, and structural elements, imagery, and forms of literary works. This is evident in the poetry of Wordsworth and Coleridge, as well as in the novels of Dickens and Hardy [8]. Within the same literary genre, writers often share commonalities in literary expression, which are frequently predicated on a shared geographical environment. For instance, the 19th-century "Lake Poets" in the UK are intrinsically and naturally linked to the Lake District of Cumberland in northern England, just as the group of writers from Nanyang in 20th-century China is connected to the unique basin environment of Nanyang [8].

**2.1.2 Research from the geography perspective.** Geographers studying literary works primarily focus on reconstructing geographic space and exploring authors' spatial expressions. These scholars draw on imaginative and behavioral geography to interpret how literary texts socially reconstruct and reimagine geographic spaces [15]. Empirical research is the predominant approach in geographic literary studies, with findings indicating that regions with advantageous geographic conditions, developed economies, active cultural exchanges, and significant transport links often experience greater literary development [14]. For example, literary works centered on the Qinhuai River contribute to Nanjing's unique spatial identity as an urban tourism destination, enhancing its appeal as an experiential tourism space [16]. Similarly, Shaanxi's literary output, influenced by its role as a cultural center and influx of talented migrants, reflects strong regional characteristics [17]. Additionally, the study of poetry highlights an evolving geographic perception of Mount Lu, progressing through stages of observing and depicting the scenery, finding tranquility within it, and achieving

enlightenment through scenic perception [18]. The representation of Egypt by Florence Nightingale, as a British female battlefield nurse, contrasts sharply from that of Gustave Flaubert, a French male writer. This divergence indicates that Egypt has been re-ordered through imagined geography to better align with the political and economic needs of different eras and audiences, thereby causing different literary experiences [19].

In summary, within the framework of literary geography research, studies by scholars from literary backgrounds often remain grounded in the theoretical paradigms of traditional Chinese literature and are less internationally oriented. In contrast, research from a geographical perspective tends to be more globally integrated, frequently employing quantitative methods and increasingly emphasizing big data analysis in recent years. The study of classical Chinese literature within this framework remains in an early stage, marked by small sample sizes, a prevalence of case studies, and limited innovation in theoretical approaches.

## 2.2 Research on the spatiotemporal characteristics of place names in classical poetry

### 2.2.1 Research object.
With the advent of digital humanities, examining the spatiotemporal characteristics of classical Chinese poetry has become a central focus in literary geography. Classical Chinese poems encapsulate extensive and varied geospatial information, including author-related spatial data (e.g., origins, residences, and travels of poets [20]) as well as creation-specific spatial information that details the locations and environments where these works were composed. Additionally, the poems frequently reference specific locations, regions, and landscapes, capturing both the places directly observed by the poets and those invoked through their imaginative associations [21]. The evolving historical context and the reconstruction of space pose challenges in verifying the modern equivalents of ancient place names. For resolving uncertainties in spatial identification, common methods include consulting prior research, reviewing local historical gazetteers from various dynasties, and examining historical maps such as the *Historical Atlas of China*, edited by Tan Qixiang [22]. The spatial scale is a key consideration in analyzing the spatiotemporal aspects of classical poetry: smaller study areas introduce greater specificity in place names but also increase the complexity of accurate spatial positioning. Current research spans multiple scales, from national [22] and regional levels [23] to urban [24] and even mountain-specific [25] studies, providing diverse insights into the spatial dimensions of classical Chinese poetry.

The study of the spatial and temporal characteristics of place names in classical poetry requires a large and representative sample of works, including the complete works of individual poets and comparable works from multiple dynasties [20,22,26]. Furthermore, analyzing changes in the spatial distribution of place names necessitates a temporal division. In classical poetry research, the more detailed the temporal division, the deeper the insights into the attributes of the subject matter. For example, segmenting the creation period of Tang poetry into five distinct phases—Early Tang, Flourishing Tang, Middle Tang, Late Tang, and the Five Dynasties—provides a more nuanced understanding of the evolution of poetic features than a simpler division into just three periods (Early, Middle, and Late Tang) [1].

Research samples are primarily sourced from two categories: traditional publications and digital resources. Traditional publications include dictionaries, poetry compilations (e.g., *Complete Song Lambic Verse Collection* edited by Tang Guizhang), individual collections of poems, and historical chronicles [3,21]. The process of collecting and verifying samples from these sources is often labor-intensive and time-consuming, with a higher risk of omissions. With the digitization of ancient texts, an increasing number of researchers now turn to dynamic databases for sample collection. Digital resources encompass both static electronic

materials and dynamic databases, including ancient poetry websites (e.g., https://sou-yun.cn/), knowledge-based platforms (e.g., Chronological and Geographical Information Platform of Tang and Song Literature) [26,27], and custom-developed software (e.g., Song Ci Writers Search Software) [28]. Additionally, some researchers extract samples from user-generated content (UGC) platforms, such as Ctrip and Qyer.

**2.2.2 Research content.** The analysis of the spatial and temporal characteristics of classical poetry focuses on the authors, exploring their distribution over time and across different geographic locations [13]. First, researchers examine the spatiotemporal distribution patterns of classical poetry authors, investigating the geographic information related to their places of origin, residence, travel, creation, and the locations referenced in their works [15,26]. This analysis also includes exploring the emotional tones expressed in their poetry, with quantitative studies on the types of emotions (e.g., positive, negative, or neutral) and their distribution over time and space using dichotomous (positive/negative) or trichotomous (positive/neutral/negative) classifications [2,5,6,23,29,30].

Second, researchers analyze the spatial and temporal characteristics of the geographic information embedded in classical poetry. This involves a quantitative assessment of the spatial distribution and the spatiotemporal evolution of the landscapes depicted in the poems [3,4,23,25,31]. This approach aims to track the shifting cultural centers and their representation in poetry over time [32].

Third, the factors influencing the spatiotemporal distribution of classical poetry and its authors are explored. Researchers attribute these patterns to various elements, including changes in political status and regime [20], cultural exchanges [33], literary traditions, economic conditions, shifts in cultural centers, and patterns of population migration [28].

Finally, the influence of classical poetry on tourism is examined. This includes investigating how poetry shapes modern tourist behavior [25], contributes to the creation of tourist destination identities, influences the development of local cultural brands [34], and fosters the growth of literary-themed tourist attractions [16].

**2.2.3 Research methods.** The methods for spatiotemporal analysis of classical poetry can be categorized into two primary approaches: qualitative and quantitative. In terms of research volume, quantitative research has emerged as the dominant approach. From a temporal perspective, the field has evolved through three stages: an initial focus on qualitative research, a subsequent integration of both qualitative and quantitative methods, and, more recently, a shift towards a predominance of quantitative analysis. Qualitative research mainly explores the temporal and spatial structural patterns of poetry, often through the lens of spatial imagery [13]. In contrast, quantitative research began with basic statistical techniques [33] but has progressively incorporated interdisciplinary methods. This includes big data analysis, visual analytics, and mathematical modeling. Key developments in the field involve the use of Geographic Information Systems (GIS) to perform spatial clustering analyses of central cultural cities [2,3,5,6,21,22,24,32], the construction of mathematical models for further analysis [4,25,29,30], and the application of software tools like Gephi and Ucinet for social network analysis [2]. Additionally, tools such as ROSTCM6 are employed to analyze the emotional content of poetry [3].

In summary, classical poetry stands as a cornerstone of China's rich traditional culture, and its study holds significant contemporary value for both cultural inheritance and cultural heritage utilization. However, there remains a notable gap in research focusing on micro-spatial scales within this field. China is home to numerous historically and culturally important cities, where prominent literary figures have created a vast body of high-quality poetic works. These works have played a pivotal role in shaping, evolving, and disseminating the cultural lineage of these cities. When studying a city, if research is confined to a single dynasty, it becomes

challenging to uncover the deeper, long-term factors influencing the formation of its cultural context. Therefore, it is crucial to employ interdisciplinary methodologies that span broader temporal periods and incorporate a sufficient number of samples. Such an approach enables a more comprehensive understanding of the city's cultural identity, its contextual foundations, and the patterns of its cultural evolution. The rise of quantitative research—incorporating geospatial data analysis, mathematical modeling, and social network analysis—has become a significant trend in literary geography. By involving scholars from diverse disciplines and applying a variety of analytical techniques, literary geography research can offer greater innovation and practical relevance.

### 2.3 Urban tourists

Urban tourists constitute a significant category within the broader spectrum of travelers, and their perceptions, preferences, and satisfaction evaluations of urban destinations have become a focal point of academic inquiry. The elements and mechanisms of tourist perception are essential for research on tourist destination perception. Through factor analysis and inductive analysis of destination perception elements, scholars have categorized the structure of destination perception into multiple dimensions, including natural resources, humanistic resources, hospitality services, and social environment [35–37]. In addition to traditional research methods such as questionnaire survey and in-depth interview, new technologies like big data and artificial intelligence have been used to enhance sample size and reduce sample bias. User-generated content (UGC) serves as a projection of tourists' authentic experiences during their activities at a destination, providing a more direct reflection of their perceptions of the urban image of the tourism site [38]. Analyzing the behaviors of a large user base provides a more comprehensive understanding of tourists' needs and preferences. This has established a new research paradigm for exploring tourist destination perceptions [39]. The use of big data-based AI-generated content (AIGC) supports and validates studies on tourists' satisfaction with destinations [40,41]. Moreover, the emotional connotations implied by the online public opinions can be used to assess the degree of positive and negative emotions expressed by tourists [42].

## 3. Materials and methods

### 3.1 Data sources and processing

This study focuses on Tang and Song poetry, which are widely regarded as the zenith of classical Chinese literature, distinguished by their substantial volume and exceptional quality. The primary textual data for this research is sourced from the Chronological Map of Tang and Song Dynasty Literature (https://cnkgraph.com/Map/PoetLife), developed by Wang Zhaopeng and his academic team at South-Central Minzu University. This digital platform, which integrates historical maps and geographic information systems (GIS), compiles biographical and literary data on 158 poets from the Tang and Song Dynasties. For the case study of Nanjing, the research draws from a collection of 1,248 poems and 97 ci poems available on the platform, including works that either describe Nanjing or were written in the city. The study specifically examines the spatial and temporal distribution of Nanjing place names within these texts, manually excluding works that do not reference specific locations within Nanjing. A total of 209 poems are retained for analysis, involving 205 distinct place names and 34 poets (13 from the Tang Dynasty and 21 from the Song Dynasty).

Due to the extensive historical timeline and the limited availability of historical records, as well as the challenges posed by later interpretations of place names, localizing ancient and historical toponyms at the urban micro-scale is particularly difficult, especially when compared

to national or provincial spatial scales. While Nanjing has retained numerous place names throughout its urban evolution, the loss of many ancient names is an inevitable consequence of this long-term transformation. Given that the Tang and Song dynasties are from a quite ancient period, the places mentioned in Tang and Song poetry have experienced more substantial changes over time. In some cases, these place names have been misinterpreted by later generations, leading to spatial shifts, with different locations sharing the same name. For example, according to the research conducted by Professor Zhang Xuefeng and others from Nanjing University [43], the Taicheng section of the Ming Dynasty City Wall today does not correspond to the Taicheng of Jiankang during the Six Dynasties. During that period, Taicheng extended from Changbai Street in the east to Yanling Lane in the west, and from Mafu Street in the south to Yangtze River Back Street in the north. Similarly, Fanglin Garden (also known as Hualin Garden) during the Six Dynasties was located within the area bounded by present-day Zhujiang Road, Taiping North Road, Longpan Road, and Beijing East Road. The Changhe Gate of Jiankang's Palace City, situated in the Six Dynasties, is now located at the southern entrance of the Baicaiyuan Residence Community on Wenchang Lane. The region between the southern bank of the inner Qinhuai River and Yuhuatai was collectively known as Changganli, subdivided into Dachanggan, Xiaochanggan, and Dongchanggan. Xuanyang Gate was positioned near the Wawa Bridge, at present-day Mafu West Street. The Zhenglu Pavilion (Baixia Pavilion) was located southwest of the Gaoqiaomen Transportation Hub in East Nanjing, on the west bank of the Qinhuai River. During the Wu period, there were at least three palaces: Taichu Palace, located in the vicinity of Zhongshan East Road, South Chaozhi Lane, and West Citang Lane; The South Place and Prince's Palace in the Yanggongjing area; and Zhaoming Palace, situated south of the current Nanjing Library. Today's Yanzhi Well is far from the location of the Six Dynasties' Jingyang Well, which was erroneously attributed to it after the Song Dynasty. Shuiting Pavilion was near the newly opened Changgan Gate, along the north bank of the Yangwu City Moat. The western city wall of Jiankang City did not extend to the area of present-day Stone City Park, and Jiankang had no outer city walls, only hedge gates. The western hedge gate, situated in the area of today's Wangfu Street, east of Chaotian Palace, marked the boundary of the western city wall. The Sunchu Restaurant Building, a name from the Tang Dynasty, is believed to have been located at the southern end of Mochou Road, near present-day Shuiximen Street, as the Mochou Lake area was still part of the river at that time. Muxiu Pavilion, historically linked with the burial of Qin Hui, was later found to be in the Jiangnan area, near Guili, based on archaeological excavations, not in the location previously assumed. The Fugui Building is near the southeast corner of today's Nanjing Ming Dynasty City Wall. Jiuhua Mountain corresponds to the current Small Jiuhua Mountain within the city, which was known as Fuzhou Mountain from the Six Dynasties to the Ming and Qing periods, and was renamed Jiuhua Mountain during the Republic of China. Wucheng is located around the present-day Gaoqiaomen Transportation Hub in eastern Nanjing. There is currently no evidence supporting the claim that the present-day Wuyi Lane corresponds to the Wuyi Lane of the Six Dynasties.

The locations of place names mentioned in Tang and Song poetry are first input into the Baidu Map API (https://api.map.baidu.com/lbsapi/getpoint/index.html) to retrieve their latitude and longitude coordinates. These coordinates are subsequently imported into ArcGIS software to analyze spatial distribution, identify hotspots, and delineate areas of concentrated activity. In this paper, emotions are categorized into three types: positive, neutral, and negative. Based on the context of each work, emotions are assigned to the corresponding locations. The spatial distribution and density of these emotional expressions are then visualized by marking the geographical coordinates on the map.

This study selects travelogues published by tourists on Ctrip as the data source for contemporary tourist spatial mobility patterns. Ctrip is one of the largest online travel service

platforms in China, with nearly 800,000 travelogues published in its travel guide section. The online travelogues collected for this study were published between January 2020 and April 2024. Using web scraping technology, over 1,284 original samples were obtained, from which 1,019 were selected for statistical analysis. By analyzing the geographic coordinates of the attractions mentioned in the travelogues, spatial visualization of the place names was conducted and compared with spatial hotspots of place names from the Tang and Song dynasties.

## 3.2 Research methods

**3.2.1 Spatial clustering analysis.** Kernel Density Estimation (KDE) is a widely utilized technique for clustering analysis, serving as an attribute-weighted measure of point-area dispersion that enables the visualization and quantitative assessment of spatial trends in research subjects. It is particularly effective in exploring the regional distribution and evolutionary patterns of locations [22]. The HDBSCAN algorithm, a density-based clustering method grounded in hierarchical clustering, employs variable mutual reach distances to distinguish clusters of varying densities from sparse noise points, providing greater precision compared to the DBSCAN algorithm [44]. As a result, HDBSCAN is frequently applied in the clustering analysis of tourism hotspots. This study employs the HDBSCAN algorithm to analyze the hotspot areas in Nanjing mentioned in Tang and Song poetry. The specific place names referenced in these works are typically represented as points in geospatial space. By examining the distribution characteristics of these points and conducting clustering analysis, this study seeks to gain insights into the spatial perceptions and cultural sentiments of Nanjing among travelers (i.e., the poetry authors) during the Tang and Song dynasties. This approach further contributes to understanding the development of Nanjing's historical and cultural lineage.

**3.2.2 Spatial overlay analysis.** This study examines the evolution of tourist spatial hotspots in Nanjing, both historically and in the present day, with a focus on the extent of overlap between these hotspots. It explores the influence of the city's historical culture on contemporary tourist behavior by analyzing the overlap of Points of Interest (POI) derived from Tang-Song poetry and online travelogues. The study employs the "center of gravity" (CoG), also known as the mean center, as an indicator of spatial distribution. POIs are calculated through the weighted average of coordinates within a given area, reflecting the spatiotemporal trajectory of spatial activities. The overlap analysis is conducted from two perspectives: the range of the standard deviational ellipse (SDE) and the CoG distribution. SDE, a spatial statistical method, effectively captures key aspects of spatial distribution, including centrality, dispersion, orientation, and spatial morphology.

**3.2.3 Emotional type analysis.** Poetry was the primary means for ancient Chinese literati to express their emotions, with different poems expressing varied emotions of their authors. There are differences in poetic styles across different eras. The fundamental characteristics of Tang poetry are its metrical structure and technical skill, along with its subtle and implicit imagery. In contrast, the basic features of Song poetry are straightforwardness, accessibility, and catchy quality, employing a direct expression of feelings rather than subtlety and ambiguity [45]. Scholars typically use a dichotomous approach (positive emotions and negative emotions) and a trichotomous approach (positive emotions, neutral emotions, and negative emotions) when analyzing the emotions expressed in classical poetry. The trichotomous approach is more widely used due to its mode detailed classification. Compared to data such as time and location, the emotional information of a work is more challenging to obtain, as the emotions expressed are often closely related to the poet's specific temporal and spatial context, as well as the events they experienced. Moreover, the complexity of classical Chinese grammar, coupled with numerous ambiguities in the definition of emotional

semantics, renders methods based on emotional dictionaries and machine learning inadequate for sentiment analysis of ancient Chinese texts [2]. Therefore, this study uses a manual annotation approach to analyze each sample of Tang and Song poetry related to Nanjing, focusing on aspects such as the creation background, writing techniques, word choices, and thematic implications. The emotions associated with the locations mentioned in the poetry are categorized into three types: positive emotions, neutral emotions, and negative emotions.

## 4. Results

### 4.1 Spatiotemporal distribution characteristics of place names

#### 4.1.1 Temporal distribution characteristics.

**(1) Temporal variations in the frequency of place mentions**

In Tang-Song poetry, the ten most frequently mentioned locations in Nanjing are: Purple Mountain, Stone City, Bailuzhou, Fenghuangtai, Xuanwu Lake, Xinting, Dongshan, Taicheng, Sanshan, and Shangxin Pavilion (Table 1). This study primarily uses present-day place names, retaining historical names where no modern equivalent exists (e.g., Xingting). A comparative analysis of the Tang and Song dynasties reveals that most of these locations were referenced in both periods, with a notable increase in frequency during the Song dynasty. This suggests that many of Nanjing's place names had already become key cultural symbols by the Tang and Song eras. Several of these names remain in use today, forming an integral part of Nanjing's cultural heritage and have since been developed into prominent tourist attractions.

**(2) Emotional trends across historical periods**

The analysis and summary of the creation background and content of Tang and Song poetry related to Nanjing reveal that positive emotions occur 29 times, neutral emotions 159 times, and negative emotions 120 times. Neutral emotions constitute the largest proportion, at 51.6%, followed by negative emotions at 39%, and positive emotions, the least, at just 9.4% (Table 2). As the "Ancient Capital of the Six Dynasties" and the "Capital City for Ten Dynasties", Nanjing has experienced numerous dynastic rises and falls, naturally leading poets to express feelings of lament and reflection. Consequently, expressions of positive emotions are rare. From a comparative perspective of the Tang and Song dynasties, travel activities during the Song dynasty were more frequent, resulting in a noticeable increase in poetry related to Nanjing. While neutral and negative emotions increased in the Song dynasty compared to the

**Table 1. Frequency of Place Mentions Across Different Periods (Top Ten).**

| Place Names | Total Frequency | Frequency in the Tang Dynasty | Frequency in the Song Dynasty |
|---|---|---|---|
| Purple Mountain | 28 | 9 | 19 |
| Stone City | 21 | 7 | 14 |
| Bailuzhou | 16 | 5 | 11 |
| Fenghuangtai | 10 | 5 | 5 |
| Xuanwu Lake | 9 | 6 | 3 |
| Xinting | 8 | 4 | 4 |
| Dongshan | 8 | 5 | 3 |
| Taicheng | 8 | 4 | 4 |
| Sanshan | 8 | 4 | 4 |
| Shangxin Pavilion | 8 | 0 | 8 |

**Table 2. Frequency of Emotions Across Different Periods (Top Ten).**

| Emotions | Total Proportion (%) | Proportion of the Tang Dynasty (%) | Proportion of the Song Dynasty (%) |
|---|---|---|---|
| Positive | 9.4 | 17.4 | 5.0 |
| Neutral | 51.6 | 42.2 | 56.8 |
| Negative | 39.0 | 40.4 | 38.2 |

Tang dynasty, the occurrence of positive emotions decreased. This shift suggests a growing concern among poets for the nation and its people, reflecting broader changes in national strength between the Tang and Song eras. Additionally, Nanjing's rich historical legacy and frequent dynastic transitions further influenced the emotional responses of poets.

**4.1.2 Characteristics of spatial distribution.** Classical poetry is rich in spatial perception and geographic imagination, capturing not only the poet's intuitive sense of urban space at the time of composition but also the local spatial elements and the historical spatial projection of the city in the poet's mind. Relying solely on traditional literary analysis makes it challenging to reveal the spatial patterns embedded in poetic works that relate to historical spaces from specific periods. Therefore, the application of modern geospatial analysis tools is crucial for examining the clustering, spatial differentiation, and temporal evolution of poets' spatial perceptions. This study employs GIS tools to explore the spatial and temporal patterns of place names referenced in Chinese poetry across different historical periods.

The location hotspots referenced in Tang and Song poetry exhibit distinct spatial patterns between the main urban areas and the suburbs of Nanjing. In the main urban area, these hotspots are concentrated in clusters, while in the suburbs, they are more dispersed (Fig 1a). The locations mentioned in Tang poetry are relatively few, with minimal clustering, whereas Song poetry shows a significant increase in the number and concentration of locations. This shift suggests a southward movement of the economic center from the Tang to the Song Dynasty, leading to a greater concentration of Nanjing's locations in Song-era poetry. From a spatial perspective, the number of locations referenced in Song Dynasty poetry surpasses those in Tang poetry, indicating a broader scope of activity by Song people in Nanjing.

Furthermore, the locations mentioned in Song poetry are predominantly concentrated in the southern parts of the city, a trend that can be attributed to several factors: the more rapid economic development in the south compared to the north, the geographical barrier of the Yangtze River limiting north-south communication, and the political tensions following the Jingkang Incident. The spatial perceptions reflected in Tang and Song poetry also reveal differences. In the Tang Dynasty (Fig 1b), the most frequently mentioned area is the vicinity of the Six Dynasties capital, the economic, political, and cultural hub of the time. This is followed by the areas of Purple Mountain and Qixia Mountain, which were cultural centers for Buddhist and Chu traditions. Other notable mentions include the western Yuhuatai area and the Dongshan region in Jiangning, associated with leisure and retreat. Spatially, these locations stretch from Xuanwu Lake in the north, westward to the Yangtze River, south to Changganli, and east to Purple Mountain. In the Song Dynasty (Fig 1c), the frequency of mentions of Purple Mountain and its surrounding areas significantly increased, altering the spatial focus that was centered on the Six Dynasties palace city during the Tang period. The mention of locations along the Yangtze River also grew. Notably, influenced by poets like Yang Wanli, who were active around Shijiu Lake, the Lishui and Gaochun areas saw a marked rise in mentions. Conversely, as economic activities shifted south, mentions of the northeastern regions diminished, while the southern part of the inner Qinhuai River emerged as a new focal point, replacing the Qixia Mountain area as a key site of mention.

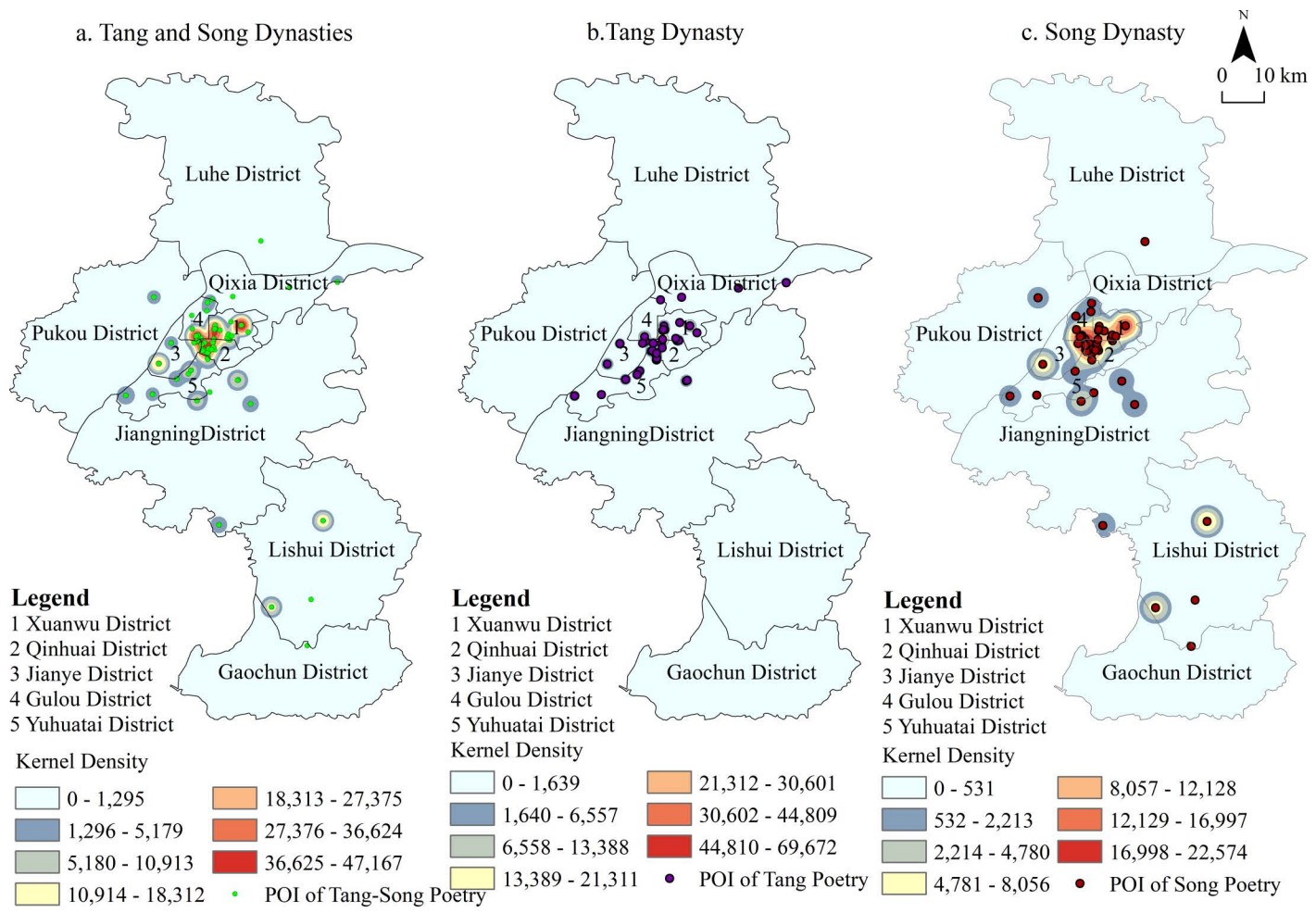

**Fig 1. POI and Kernel Density of Locations Mentioned in Tang-Song Poetry.**

**4.1.3 Emotional distribution patterns and characteristics.** Figs 2–4 depict the spatiotemporal distribution of emotions associated with the places mentioned in the Tang and Song dynasties' poetic works. Each dot represents a location, with color coding indicating the type of emotion conveyed: green for positive emotions, blue for neutral emotions, and purple for negative emotions. By analyzing the emotional data spatiotemporally, the figure provides insights into the emotional responses of poets (travelers) toward various locations in Nanjing during these two dynasties. It also reveals the temporal and spatial distribution of these emotions, both within individual periods and across different time frames. Overall, the poems from these eras convey fewer positive emotions and more negative and neutral sentiments toward Nanjing's locations (Fig 2). The density clustering distribution further highlights the spatial characteristics of these emotions: positive emotions are predominantly concentrated in the Xuanwu District, while negative emotions are more prevalent in the Qinhuai and Gulou Districts. Neutral emotions are notably distributed across Xuanwu and Qinhuai Districts, with Jianye District emerging as a secondary center for neutral emotions.

The spatial distribution of travelers' (or poets') emotions towards Nanjing during the Tang and Song dynasties reveals distinct patterns. Overall, the clustering of all three emotional types is more pronounced in the Song Dynasty compared to the Tang Dynasty (Figs 3 and 4).

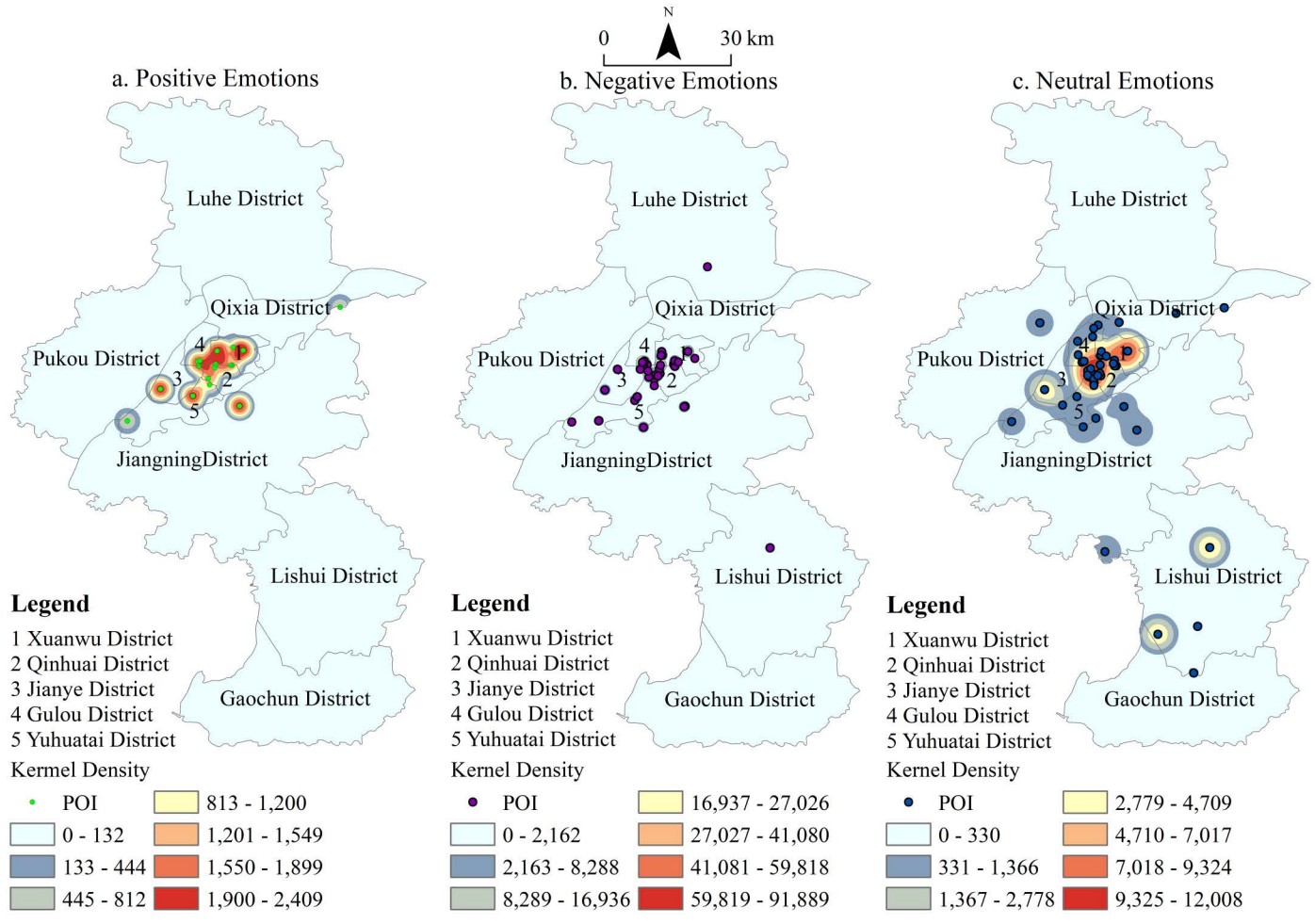

**Fig 2. POI and Kernel Density of Emotions Related to Place Names in Tang-Song Poetry.**

Negative emotions in the Tang Dynasty were primarily concentrated in the palace city of the Six Dynasties. However, in the Song Dynasty, these emotions expanded southwestward, creating three prominent hotspots: the Six Dynasties Palace City, Stone City, and the Old South City area. Positive emotions, which were less widespread in the Tang Dynasty, grew notably in the Song Dynasty, with a concentration in the southern part of Xuanwu Lake. In the Tang Dynasty, neutral emotions were concentrated in two main areas: the Six Dynasties Palace City and Purple Mountain. By the Song Dynasty, the distribution of neutral emotions expanded, becoming more concentrated in the Six Dynasties Palace City, Xuanwu Lake, Purple Mountain, and Stone City. Density clustering analysis reveals that the frequent changes in dynasties evoked strong emotional responses from the literati, shaping a distinctive cultural perception of Nanjing that reflects the city's historical vicissitudes.

## 4.2 Overlay analysis of spatial clustering of Nanjing place names between ancient and modern times

### 4.2.1 Spatial distribution of Nanjing's place names in online travel narratives.
This study is based on travelogues published by tourists on Ctrip and employs the open API

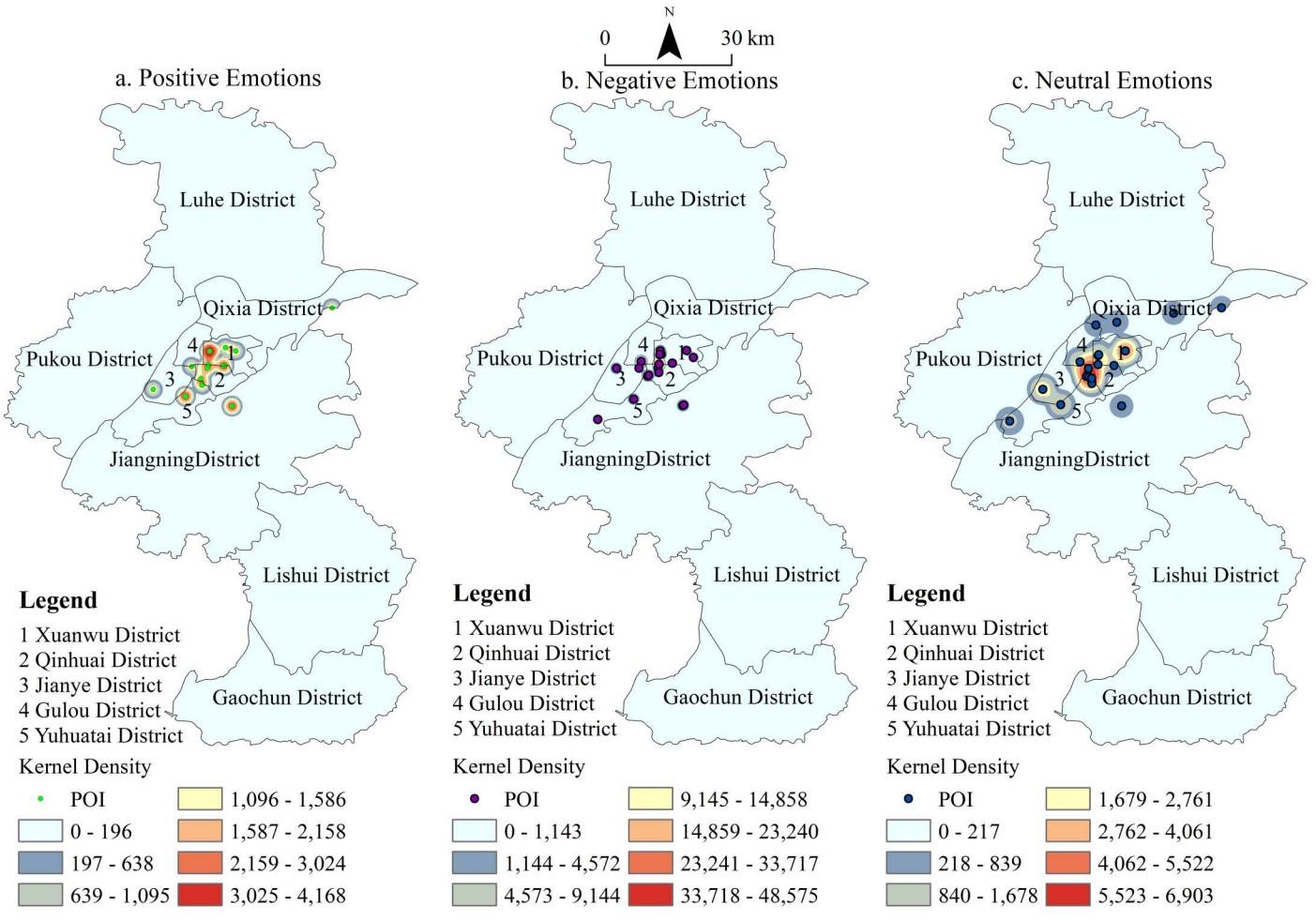

**Fig 3. POI and Kernel Density of Emotions Related to Place Names in Tang Poetry.**

platform of Baidu Maps to obtain the latitude and longitude of the attractions mentioned in the travelogues. Using this information, the spatial distribution of place names and clustering hotspots were mapped (Fig 5). Most tourist attractions are concentrated in the Xuanwu, Qinhuai, and Gulou districts—areas that make up the historical core of Nanjing. This concentration reflects the city's rich cultural heritage and aligns with its status as a historical and cultural hub. Notably, the locations frequently mentioned in the travelogues, such as Xuanwu Lake, Purple Mountain, Qinhuai River, Jiming Temple, Wuyi Lane, Niushou Mountain, Mochou Lake, and Qixia Mountain (Temple), overlap with those cited in Tang and Song poetry. Kernel density analysis reveals density centers in areas around the inner Qinhuai River, the southern bank of Xuanwu Lake, and the vicinity of Purple Mountain, which correspond to density levels four, three, and two, respectively. These density hotspots are largely consistent with those identified in Tang-Song poetry.

**4.2.2 Overlay analysis of spatial clustering of place names in Tang-Song poetry and online travelogues.** From the perspective of Spatial Data Exploration (SDE) (Fig 6a, b), the points of interest (POIs) referenced in Tang and Song poetry are primarily concentrated in the southern part of central Nanjing, displaying a long and narrow distribution. These POIs encompass

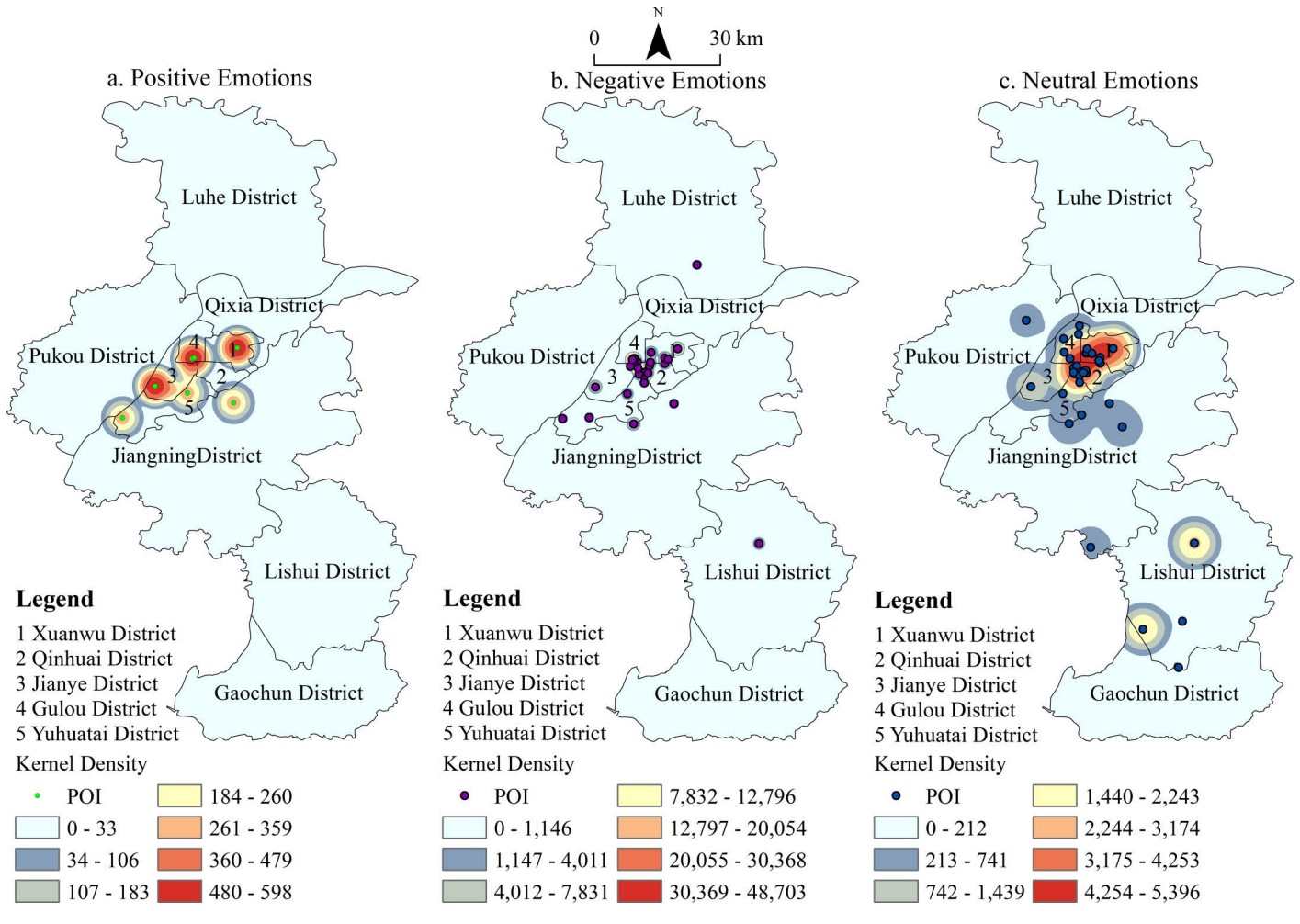

**Fig 4. POI and Kernel Density of Emotions Related to Place Names in Song Poetry.**

much of Nanjing's main urban area, as well as parts of Qixia, Pukou, and Jiangning Districts. In contrast, the POIs mentioned in online travelogues are more concentrated in the northern part of the city, with a narrower distribution compared to those in Tang and Song poetry. This shift suggests a more pronounced clustering of locations within the main urban area in contemporary travelogues, in contrast to the broader spread seen in historical poetry. Notably, the districts of Gaochun, Lishui, Luhe, and Pukou fall outside the SDEs of both the Tang-Song poetry POIs and those in online travelogues, highlighting a distinct core-periphery structural pattern in Nanjing's POIs from ancient times to the present. This pattern is primarily attributable to the main urban area's over 3,100-year history and approximately 450 years as a capital, which has endowed it with rich cultural and tourism resources. Additionally, the two 5A-rated scenic areas in Nanjing—the Confucius Temple-Qinhuai Scenic Area and the Zhongshan Scenic Area— are both situated within the main urban area, alongside numerous other popular tourist sites, contributing to the heightened attention and significance of this central region.

From the perspective of center of gravity (CoG) distribution (Fig 6a,b), the CoG of points of interest (POIs) mentioned in Tang and Song poetry is located at coordinates (118°48′36″E, 31°59′20″N), on the edge of the Qinhuai District, near the Jiangning District. In contrast, the CoG of the POIs referenced in online travelogues is situated at (118°48′47″E, 32°1′48″N),

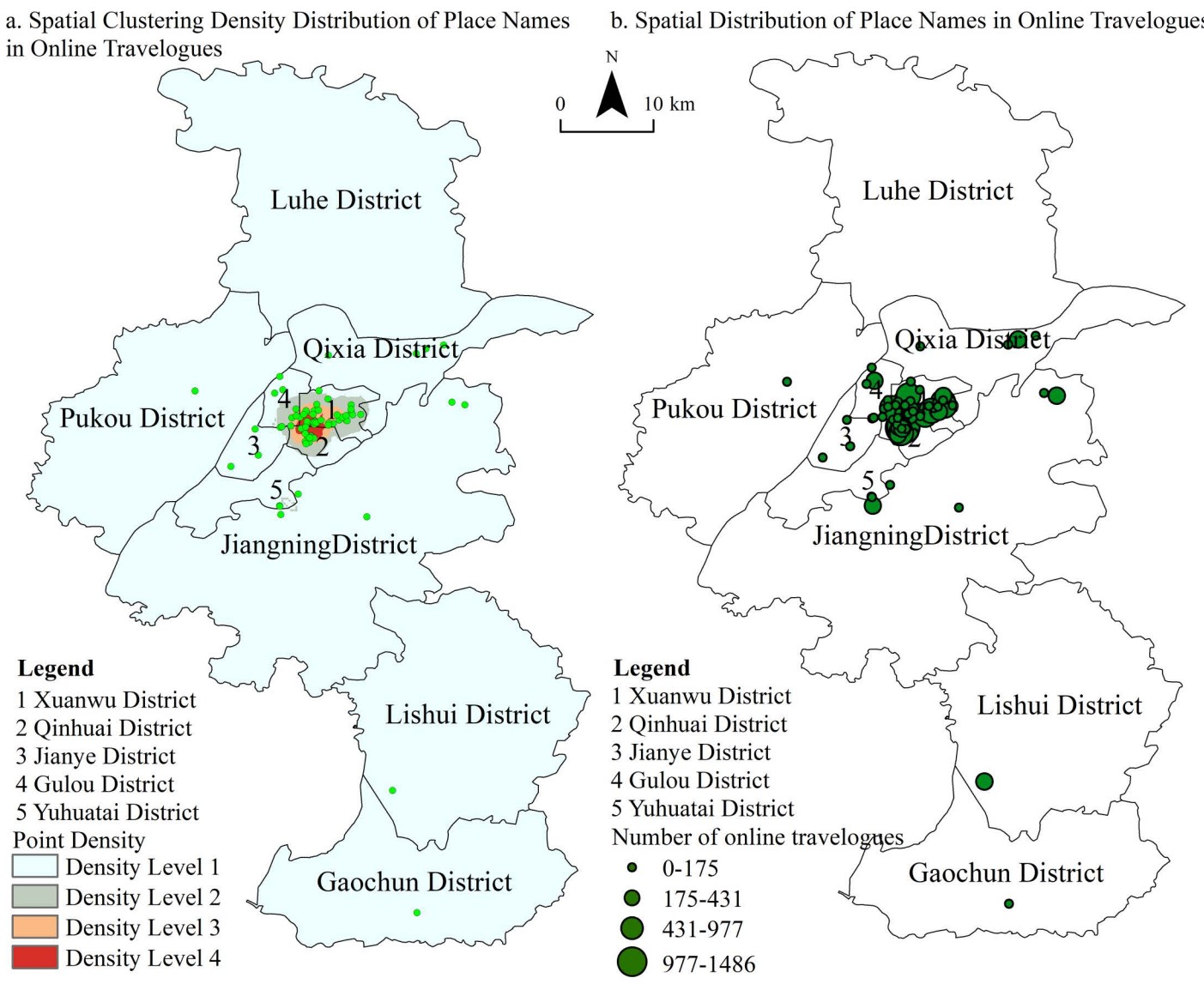

**Fig 5. POI and Kernel Density of Nanjing Place Names in Online Travelogues.**

slightly northeast of the POIs in Tang and Song poetry, in the central area of the Qinhuai District, close to Xuanwu District. The distance between the two CoGs is 4.59 km, indicating that the CoGs of the locations mentioned in both Tang-Song poetry and online travelogues are only marginally different, with minimal overall shift. This suggests that the main urban area continues to be the dominant cultural and tourism hub of Nanjing. The slight northeastward shift of the CoG for online travelogues can likely be attributed to the increased appeal of the central urban area's resources, transportation networks, and commercial advantages, particularly during peak tourist seasons, resulting in a more pronounced clustering effect.

An analysis of the overlap between POIs mentioned in Tang-Song poetry and those in online travelogues (Fig 6c) shows that, with the exception of one overlapping point in Lishui District, most POIs are located within the main urban area. A kernel density analysis of these overlapping points indicates that the highest density region is situated in Xuanwu District. With its two

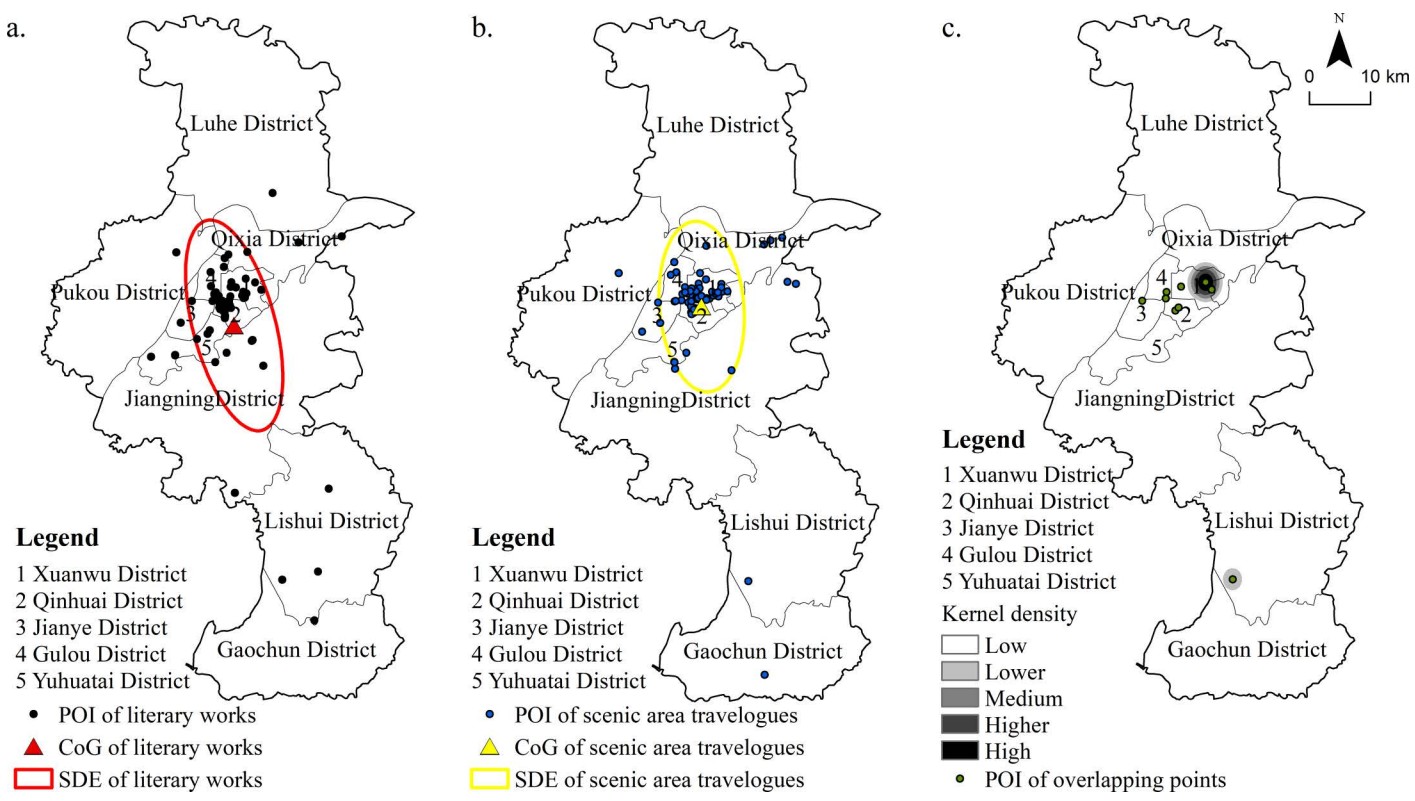

**Fig 6. Comparison of Spatial Distribution of Locations Mentioned in Tang-Song Poetry and Online Travelogues.**

prominent attractions, Xuanwu Lake and the Zhongshan Scenic Area, Xuanwu District stands out as the most significant POI in both Tang-Song poetry and contemporary travelogues.

## 5. Conclusion and discussion

This study explores the temporal and spatial changes in the locations referenced in Tang-Song poetry by collecting and verifying place names and analyzing their spatial distribution. The paper also compares the distribution of these locations in Tang and Song poetry with those found in contemporary online travelogues, focusing on the same geographical context of Nanjing. The following conclusions can be drawn:

First, there is a continued emphasis on the same locations in both Tang and Song poetry, a cultural tradition that persists to the present day. As a result, these locations have evolved into significant tourist attractions and cultural landmarks. Second, the spatial hotspots identified in Tang and Song poetry reveal a clustered distribution in the main urban areas of Nanjing, with a more scattered pattern in the suburbs. Compared to the Tang Dynasty, Nanjing's urban appeal grew significantly during the Song Dynasty, broadening the scope of activities for both tourists and poets, and stimulating economic development. The shift in these activity hotspots is closely linked to broader economic, political, and cultural factors. Third, when comparing the Tang and Song periods, there is a notable increase in neutral and negative emotions and a decrease in positive emotions in Song Dynasty poetry. This shift reflects the overall decline in national strength during this period, with Nanjing evoking feelings of reflection and melancholy due to its rich historical heritage and frequent dynastic transitions. Fourth, when comparing the spatial distribution from the Tang and Song periods to contemporary times, the

spatial hotspots show a broader geographical spread, with a slight shift of focus to the north. Despite these changes, key points of interest (POIs), such as Xuanwu Lake and the Zhongshan Scenic Area, remain consistently featured in both Tang and Song poetry and modern travelogues, highlighting the enduring significance of these cultural sites. The clustering of these POIs is more pronounced, suggesting a core-periphery structure.

This study employs GIS-based spatial statistical methods to provide a collective, quantitative, and visualized analysis of the spatial cognition of poets from the Tang and Song dynasties, as well as contemporary tourists, through the examination of poetry and online travelogues. The research contributes to the fields of literary geography and literary tourism at the urban spatial scale, offering new insights into the cultural continuity of cities. By comparing the evolving place identity among historical and contemporary urban travelers, this study presents a novel perspective on the cultural legacy of cities.

However, limitations exist within the scope of this research. First, the focus is restricted to the Tang and Song dynasties, a relatively short period in China's long history. Second, the case study is limited to Nanjing, despite the existence of numerous other historical and cultural cities in China. Expanding the sample to include additional cities would provide more comprehensive insights. Third, in selecting samples, only poetry that explicitly mentions place names were included in the analysis, while works that refer to cultural sites indirectly were excluded. Future research should seek solutions to address this limitation. Finally, while the study analyzes classical poetry and modern online travelogues, literary works extend beyond these genres. Ancient prose and contemporary literature also offer valuable data for textual analysis, and this represents an area for future exploration.

## Supporting information

**S1 File. Information on Tang and Song Literature Works in Nanjing.**
(XLSX)

**S2 File. Online travelogues information from Ctrip.**
(XLSX)

## Author contributions

**Conceptualization:** Weiya Zhang, Shu Tang.

**Data curation:** Weiya Zhang.

**Formal analysis:** Shu Tang.

**Funding acquisition:** Weiya Zhang.

**Investigation:** Zhicheng Lai.

**Methodology:** Weiya Zhang, Zhicheng Lai.

**Resources:** Shu Tang.

**Writing – original draft:** Weiya Zhang, Zhicheng Lai.

**Writing – review & editing:** Shu Tang.

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
