## [Decision Letter · Decision Letter 0]

3 Jan 2025

Dear Dr. Tang,

Thank you for submitting your manuscript to PLOS ONE. After careful consideration, we feel that it has merit but does not fully meet PLOS ONE’s publication criteria as it currently stands. Therefore, we invite you to submit a revised version of the manuscript that addresses the points raised during the review process.

We look forward to receiving your revised manuscript.

Kind regards,

Dawit Dibekulu, PhD

Academic Editor

PLOS ONE

Journal Requirements:

“This work was supported by the Ministry of Education Humanities and Social Sciences Foundation Project (Grant No. 24YJA790090).”

“This work was supported by the Ministry of Education Humanities and Social Sciences Foundation Project (Grant No. 24YJA790090,).We also thank the anonymous reviewers for their valuable comments, which greatly improved the manuscript.”

“This work was supported by the Ministry of Education Humanities and Social Sciences Foundation Project (Grant No. 24YJA790090).”

5. We note that your Data Availability Statement is currently as follows: All relevant data are within the manuscript and its Supporting Information files.

6. PLOS requires an ORCID iD for the corresponding author in Editorial Manager on papers submitted after December 6th, 2016. Please ensure that you have an ORCID iD and that it is validated in Editorial Manager. To do this, go to ‘Update my Information’ (in the upper left-hand corner of the main menu), and click on the Fetch/Validate link next to the ORCID field. This will take you to the ORCID site and allow you to create a new iD or authenticate a pre-existing iD in Editorial Manager.

7. We note you have included a table to which you do not refer in the text of your manuscript. Please ensure that you refer to Table 2 in your text; if accepted, production will need this reference to link the reader to the Table.

Reviewers' comments:

Reviewer's Responses to Questions

**Comments to the Author**

1. Is the manuscript technically sound, and do the data support the conclusions?

Reviewer #1: Yes

Reviewer #2: Yes

2. Has the statistical analysis been performed appropriately and rigorously?

Reviewer #1: Yes

Reviewer #2: Yes

3. Have the authors made all data underlying the findings in their manuscript fully available?

Reviewer #1: Yes

Reviewer #2: No

4. Is the manuscript presented in an intelligible fashion and written in standard English?

Reviewer #1: Yes

Reviewer #2: Yes

Reviewer #1: The research gap presented in the introduction is not sufficiently clear. Please provide a more detailed explanation of the existing gap and how your proposed research will address it."In addition to this it is very important to clarify the contribution of your study to the international community.

Reviewer #2: Investigating literary works from a geographic perspective, this study analyzed the spatial and temporal distribution of place names in 209 Tang-Song poems and compared them to those in online travel narratives. There were four main findings. First, the study reported differences in cultural hotspots featured in Tang poetry and Song poetry. Second, there were more locations mentioned in Song poetry than Tang poetry. Third, a sentiment analysis revealed a shift towards neutral and negative emotions in Song poetry than Tang poetry. Fourth, some cultural hotspots remained consistently referenced in classical poetry and modern travel narratives. Overall, the study has a clear motivation, a strong design and is well-written. I have only a few comments that I wish the authors to address to enhance the current manuscript. Therefore, I recommend minor revision.

Main comments

Motivation

1) The authors provide a thorough explanation of the research gap. They highlighted that previous studies of spatiotemporal distributions have rarely approached the topic from a geographic perspective using extensive longitudinal data. However, the manuscript's comparative analysis between historical poetry and modern travel narratives would benefit from a clearer grounding. While the authors implicitly draw parallels between Tang-Song poets as travelers and contemporary tourists, this connection deserves more explicit development in the theoretical framework. The poets' dual role as both literary figures and travelers appears to be a key premise of the study, but this perspective needs to be more clearly articulated. The conclusion presents an intriguing argument about how comparing historical and contemporary urban travelers can offer novel insights into cities' cultural legacies. I wish the authors would introduce this valuable perspective earlier in the manuscript, ideally in the introduction, to better motivate the study.

Literature review

1) Please briefly review prior studies examining contemporary urban travelers. This can allow the authors to highlight the lack of studies comparing historical and modern travelers.

2) While the Chinese literary geography coverage is strong, consider briefly discussing similar work from other regions (UK, US, etc.) to provide international context.

Methods

1) The study analyzed classical poetry as well as contemporary travel narratives. However, in the data sources and processing section, the study discusses in detail the collection of classical poetry and only mentions how modern travel narratives were collected on page 15. Please consider moving this discussion to the data preprocessing/methods section.

2) Briefly explain the process of excluding works that do not reference specific locations in Tang-Song poetry. Was this automated or manual?

3) P.8: The manuscript mentions that some places have undergone a spatial shift overtime. Please provide evidence for historical place name changes and spatial shifts.

4) P.9: regarding this part “Emotion plays a critical role in classical Chinese poetry, serving as the subjective expression of the authors' perceptions of place. This emotional dimension provides valuable insights for studies in literary geography.” I believe discussing the significance of examining emotions here may be repetitive as the manuscript previously discussed this point in the literature review and introduction.

5) Why were three categories used to code emotions in this study? This decision was only well-motivated in the “research outcomes” section. It would be better to move this motivation earlier to the methods/data preprocessing section.

6) In the frequency analysis, consider using proportions instead of raw counts for better comparability

7) Address the sample size disparity between dynasties (109 Tang poems vs 199 Song poems)

8) Please specify the emotion classification tool and its accuracy

9) Consider how poetic techniques (imagery, allusions, metaphors) may affect emotion detection, particularly given the stylistic differences between Tang and Song poetry. Some studies support the distinct linguistic expression in Tang versus Song poetry (Yang, Zheng, & Shao, 2018).

10) Account for how a poem's genre might have influenced emotional expression, and whether the genre effect may have played a role in the current findings.

Reference:

- Yang, Y., Zheng, Z., & Shao, Y. (2018). A Study of Color Words in Tang Poetry and Song Lyrics. In Chinese Lexical Semantics: 19th Workshop, CLSW 2018, Chiayi, Taiwan, May 26–28, 2018, Revised Selected Papers 19 (pp. 245-255). Springer International Publishing.

Minor comments

1) Some terms are introduced without definitions. Incorporating the intended definition for each term will enhance the readability of the paper. The terms include: cultural landscapes, digital humanities techniques, and toponyms.

2) P.3: please cite studies which lie in this group to guide interested readers to the relevant literature: “The third area of focus examines the relationship between literary genres and geography”

3) P.6: The reader may not understand the relevance of “Marxist principles” in this sentence “classical poetry stands as a cornerstone of China’s rich traditional culture, and its study holds significant contemporary value for both cultural preservation and the integration of Marxist principles with traditional Chinese heritage”. Please clarify this phrase to enhance the coherence of the paper.

4) P.8: in the sentence “Given that the Tang and Song dynasties predate the Ming and Qing dynasties, “, the comparison with the Ming and Qing may not be necessary. The sentence may focus on how long ago the Tang and Song dynasties were, which could highlight the centuries of change these cultural locations have experienced.

5) Consider mentioning in the limitations that only poems that explicitly feature names were included in the analysis, which constituted a very small sample of Tang-Song poetry, while those which only described a cultural place were not included. This limitation may guide future research to explore a technique that addresses this issue.

**Do you want your identity to be public for this peer review?** For information about this choice, including consent withdrawal, please see our Privacy Policy

Reviewer #1: **Yes: ** Dr.Addisu Alehegn

Reviewer #2: **Yes: ** Alaa Alzahrani

---

## [Author Response · Author response to Decision Letter 1]

26 Jan 2025

Response to Main Comments

Motivation

1) The authors provide a thorough explanation of the research gap. They highlighted that previous studies of spatiotemporal distributions have rarely approached the topic from a geographic perspective using extensive longitudinal data. However, the manuscript's comparative analysis between historical poetry and modern travel narratives would benefit from a clearer grounding. While the authors implicitly draw parallels between Tang-Song poets as travelers and contemporary tourists, this connection deserves more explicit development in the theoretical framework. The poets' dual role as both literary figures and travelers appears to be a key premise of the study, but this perspective needs to be more clearly articulated. The conclusion presents an intriguing argument about how comparing historical and contemporary urban travelers can offer novel insights into cities' cultural legacies. I wish the authors would introduce this valuable perspective earlier in the manuscript, ideally in the introduction, to better motivate the study.

We sincerely appreciate the reviewer's insightful observation regarding the dual roles of classical Chinese literati as both literary figures and travelers, which has profoundly informed our analytical framework. In the revised manuscript, we have explicitly articulated and substantiated this perspective through targeted textual expansions and conceptual clarifications. The comprehensive revisions addressing this perspective can be found in Lines 39-42 and Lines 75-84 of the revised manuscript.

Literature review

1) Please briefly review prior studies examining contemporary urban travelers. This can allow the authors to highlight the lack of studies comparing historical and modern travelers.

In response to your recommendation, we have added Section “2.3 Urban Tourists” (Lines 242-259) to the literature review. This enhancement has substantially strengthened the theoretical foundation of our study.

2) While the Chinese literary geography coverage is strong, consider briefly discussing similar work from other regions (UK, US, etc.) to provide international context.

We sincerely appreciate your suggestion. Our initial analysis indeed lacked a cross-cultural comparative perspective. To address this limitation, we have incorporated research findings on the relationship between British literary genres and their geographical determinants, thereby enhancing the international relevance of this study's scholarly contribution. The comprehensive revisions can be found in Lines 117-125 and Lines 139-143 of the revised manuscript.

Methods

1) The study analyzed classical poetry as well as contemporary travel narratives. However, in the data sources and processing section, the study discusses in detail the collection of classical poetry and only mentions how modern travel narratives were collected on page 15. Please consider moving this discussion to the data preprocessing/methods section.

We have moved this discussion to the “3.1 Data Sources and Processing” section. Lines 327-334.

2) Briefly explain the process of excluding works that do not reference specific locations in Tang-Song poetry. Was this automated or manual?

In the Data Processing section, we have expanded the description of our methodological protocols for cross-referencing and exclusion criteria regarding artwork information. Lines 271-273.

3) P.8: The manuscript mentions that some places have undergone a spatial shift overtime. Please provide evidence for historical place name changes and spatial shifts.

In this section, we have incorporated additional case studies, key findings, and references drawing on the authoritative scholarship of prominent Chinese historians, which we believe will substantiate the arguments presented. Lines 284-316.

4) P.9: regarding this part “Emotion plays a critical role in classical Chinese poetry, serving as the subjective expression of the authors' perceptions of place. This emotional dimension provides valuable insights for studies in literary geography.” I believe discussing the significance of examining emotions here may be repetitive as the manuscript previously discussed this point in the literature review and introduction.

We acknowledge the validity of your suggestion and have removed the redundant content in this section. Lines 320-323.

5) Why were three categories used to code emotions in this study? This decision was only well-motivated in the “research outcomes” section. It would be better to move this motivation earlier to the methods/data preprocessing section.

In Section 3.2 (Research Methods), we have added subsection 3.2.3 (Emotional Type Analysis) to provide a conceptual foundation for emotional categorization and related methodological considerations prior to discussing the results. Lines 362-381.

6) In the frequency analysis, consider using proportions instead of raw counts for better comparability.

In the frequency analysis, as shown in Table2 we have conducted proportions instead of raw counts for better comparability. Lines 414.

7) Address the sample size disparity between dynasties (109 Tang poems vs 199 Song poems).

We have provided contextualized descriptions and analytical evaluations of the historical context underlying the discrepancies in sample sizes. This enhancement aims to systematically elucidate the historical determinants contributing to such variations. Lines 400-402; Lines 406-410.

8) Please specify the emotion classification tool and its accuracy.

In Subsection 3.2.3 (Emotional Type Analysis) we have added an explanation to specify the emotion classification tool and its accuracy. Lines 377-381.

9) Consider how poetic techniques (imagery, allusions, metaphors) may affect emotion detection, particularly given the stylistic differences between Tang and Song poetry. Some studies support the distinct linguistic expression in Tang versus Song poetry (Yang, Zheng, & Shao, 2018).

In Section 3.2.3 (Emotional Type Analysis), we have expanded the discussion with necessary clarifications, addressed common classification methodologies and their application boundaries, and supplemented relevant citations to directly address your recommendation. Lines 368-377.

10) Account for how a poem's genre might have influenced emotional expression, and whether the genre effect may have played a role in the current findings.

In Section 3.2.3 (Emotional Type Analysis), we have expanded the discussion with necessary clarifications and supplemented relevant citations to directly address your recommendation. Lines 363-368.

Response to Minor Comments

1) Some terms are introduced without definitions. Incorporating the intended definition for each term will enhance the readability of the paper. The terms include: cultural landscapes, digital humanities techniques, and toponyms.

In accordance with the reviewers' suggestions, we have supplemented the definitions of key terminology in the text to facilitate readers' intuitive understanding of our conceptual framework. The specific enhancements include: Lines 44-51; Lines 65-66; Line 110-112.

2) P.3: please cite studies which lie in this group to guide interested readers to the relevant literature: “The third area of focus examines the relationship between literary genres and geography”.

In response to the reviewers' recommendations, we have expanded the reference list to enhance readers' comprehension and facilitate further exploration of the perspectives presented. Lines 115-117.

3) P.6: The reader may not understand the relevance of “Marxist principles” in this sentence “classical poetry stands as a cornerstone of China’s rich traditional culture, and its study holds significant contemporary value for both cultural preservation and the integration of Marxist principles with traditional Chinese heritage”. Please clarify this phrase to enhance the coherence of the paper.

We appreciate your suggestion as both constructive and essential to strengthening the scholarly rigor of this work. To mitigate potential misinterpretation, we have revised the relevant phrasing as follows: In summary, classical poetry stands as a cornerstone of China’s rich traditional culture, and its study holds significant contemporary value for both cultural inheritance and cultural heritage utilization. Lines 227-229.

4) P.8: in the sentence “Given that the Tang and Song dynasties predate the Ming and Qing dynasties, “, the comparison with the Ming and Qing may not be necessary. The sentence may focus on how long ago the Tang and Song dynasties were, which could highlight the centuries of change these cultural locations have experienced.

We have revised the phrasing of this sentence to read as follows: Given that the Tang and Song dynasties are from a quite ancient period, the places mentioned in Tang and Song poetry have experienced more substantial changes over time. In some cases, these place names have been misinterpreted by later generations, leading to spatial shifts, with different locations sharing the same name. Lines 281-284.

5) Consider mentioning in the limitations that only poems that explicitly feature names were included in the analysis, which constituted a very small sample of Tang-Song poetry, while those which only described a cultural place were not included. This limitation may guide future research to explore a technique that addresses this issue.

In accordance with your recommendation, we have addressed the sample size limitations in the Discussion section (Section 5) and delineated this issue as a priority research direction for future investigations. Furthermore, we have incorporated your constructive suggestion into our team's research agenda framework. We sincerely appreciate your expertise in identifying this critical methodological dimension, which will continue to guide our ongoing scholarly explorations. Lines 583-585.

---

## [Editor Report · Decision Letter 1]

30 Jan 2025

Spatiotemporal Distribution Characteristics of Nanjing Place Names — Based on Data Mining of Tang-Song Poetry and Online Travelogues

PONE-D-24-55005R1

Dear Dr. Tang,

We’re pleased to inform you that your manuscript has been judged scientifically suitable for publication and will be formally accepted for publication once it meets all outstanding technical requirements.

Kind regards,

Dawit Dibekulu, PhD

Academic Editor

PLOS ONE

Additional Editor Comments (optional):

Dear Authors ,

I am pleased to inform you that your manuscript titled has been accepted for publication. You have successfully addressed the reviewers' comments, and the paper now meets the journal's academic standards.

Thank you for your valuable contribution to the field. The editorial office will contact you shortly regarding the next steps for publication.

Congratulations on this achievement!

Warm regards,
---

## [Editor Report · Acceptance letter]

PONE-D-24-55005R1

PLOS ONE

Dear Dr. Tang,

I'm pleased to inform you that your manuscript has been deemed suitable for publication in PLOS ONE. Congratulations! Your manuscript is now being handed over to our production team.

Kind regards,

on behalf of

Dr. Dawit Dibekulu

Academic Editor

PLOS ONE